# Complementary Sets of Autoantibodies Induced by SARS-CoV-2, Adenovirus and Bacterial Antigens Cross-React with Human Blood Protein Antigens in COVID-19 Coagulopathies

**DOI:** 10.3390/ijms231911500

**Published:** 2022-09-29

**Authors:** Robert Root-Bernstein, Jack Huber, Alison Ziehl

**Affiliations:** Department of Physiology, Michigan State University, 567 Wilson Road, Room 2174 BPS, East Lansing, MI 48824, USA

**Keywords:** bystander infection, antigenic complementarity, thrombosis, thrombocytopenia, SARS-CoV-2, mRNA vaccine, adenovirus, streptococcus, staphylococcus, *E. coli*, autoimmune, autoimmunity, cross-reactive, platelet factor 4, cardiolipin, von Willebrand Factor, vaccine-induced thrombotic thrombocytopenia

## Abstract

COVID-19 patients often develop coagulopathies including microclotting, thrombotic strokes or thrombocytopenia. Autoantibodies are present against blood-related proteins including cardiolipin (CL), serum albumin (SA), platelet factor 4 (PF4), beta 2 glycoprotein 1 (β2GPI), phosphodiesterases (PDE), and coagulation factors such as Factor II, IX, X and von Willebrand factor (vWF). Different combinations of autoantibodies associate with different coagulopathies. Previous research revealed similarities between proteins with blood clotting functions and SARS-CoV-2 proteins, adenovirus, and bacterial proteins associated with moderate-to-severe COVID-19 infections. This study investigated whether polyclonal antibodies (mainly goat and rabbit) against these viruses and bacteria recognize human blood-related proteins. Antibodies against SARS-CoV-2 and adenovirus recognized vWF, PDE and PF4 and SARS-CoV-2 antibodies also recognized additional antigens. Most bacterial antibodies tested (*group A streptococci* [*GAS*], *staphylococci*, *Escherichia coli* [*E. coli*], *Klebsiella pneumoniae*, *Clostridia*, and *Mycobacterium tuberculosis*) cross-reacted with CL and PF4. while GAS antibodies also bound to F2, Factor VIII, Factor IX, and vWF, and *E. coli* antibodies to PDE. All cross-reactive interactions involved antibody-antigen binding constants smaller than 100 nM. Since most COVID-19 coagulopathy patients display autoantibodies against vWF, PDE and PF4 along with CL, combinations of viral and bacterial infections appear to be necessary to initiate their autoimmune coagulopathies.

## 1. Introduction

SARS-CoV-2 is a new coronavirus that causes symptoms ranging from minor ones such as fever, sore throat, nasal congestion or head- and muscle aches to moderate ones including dyspnea, muscle weakness and chronic loss of smell and/or taste, and severe ones including coagulopathies such as thrombocytopenia, disseminated intravascular coagulopathy (DIC), microclotting, impaired circulation, venous thromboembolisms (VTE) resulting in heart attacks and strokes, respiratory complications and other types of organ failure [1,2,3]. While mild cases of COVID-19 have no increased risk of coagulopathies [4], 10–15% of hospitalized patients [5], 25% of critically ill COVID-19 patients and up to 48% of intensive care patients [3,6,7,8,9,10] develop coagulopathies, which predominantly affect the elderly [11]. The incidence of COVID-19-related coagulopathies has been found to be about ten times the rate observed among hospitalized influenza patients [5,11,12]; fibrin structure and fibrinolysis are altered in comparison to both influenza patients and normal individuals [13]; and VTE were twice as common among COVID-19 patients as among those with community acquired pneumonias [9]. Thrombotic complications also occur in about 1 in 25,000 to 100,000 people vaccinated against SARS-CoV-2 [14,15], with a higher incidence among those receiving adenovirus vector vaccines than mRNA vaccines.

Various causes of COVID coagulopathies have been proposed including genetics, defects in the renin-angiotensin system, defective platelet gene expression, endothelitis, and cytokine storm-induced complement activation [6,16,17,18,19]. Vaccine-associated thrombosis has been suggested to be caused by the ethylenediaminetetraacetic acid (EDTA) preservative in the AstraZenaca formulation [20], but this proposal does not explain the increased risk associated with the mRNA vaccines. Another suggestion is that platelet factor 4 (PF4) binds to the vaccines creating a complex that induces novel antibodies that activate coagulation pathways [21], or that the vaccine damages the glycocalyx releasing fragmented forms of glycosaminoglycans that mimic heparin, which upon binding to PF4, trigger coagulation [22]. However, the best-documented cause for COVID-19-associated coagulopathies is autoimmune [23,24].

Autoantibodies against a wide range of blood proteins have been documented in COVID-19 patients and SARS-CoV-2 vaccinees, the targets of which include phospholipids and phospholipid-binding proteins [25]; lupus anticoagulant, cardiolipin (CL), and the cardiolipin-binding proteins phosphatidylserine/prothrombin (Factor 2), platelet glycoprotein Ib (GP1b) and beta-2 glycoprotein I (β2GPI) [7,25,26,27,28,29]; PF4 [30,31,32,33] von Willebrand factor (VWF), ADAMTS13 (von Willebrand factor-cleaving protease or VWFCP), and Factors, IX, X and Xa [34,35,36,37,38]. Some of these autoantibodies are also found transiently in many COVID-19 patients who do not develop coagulopathies and among COVID-19 vaccinees, which challenges whether SARS-CoV-2 or its vaccines are sufficient to induce autoimmune coagulopathies [24,27,29,32,33]. Depending on the cut-off values used, between 30 and 75% of people vaccinated against SARS-CoV-2 develop autoantibodies against PF4 but do not display any coagulopathy symptoms [29,32,33]. Thus, one challenge is to explain why the vast majority of people infected with or vaccinated against SARS-CoV-2 fail to develop autoimmune coagulopathies and why the presence of autoantibodies may not translate directly into active autoimmune disease. A related challenge is to explain why patients develop different coagulopathies ranging from thrombocytopenia to disseminated microclotting to VTE.

The antigens targeted in COVID-19-associated autoimmune coagulopathies generally fall into two broad categories composed of clotting factors and platelet proteins (Table 1). The main clotting factor antigens are Factors 2, VIII, IX, X, VWF and ADAMTS13, all of which are involved in a complex cascade of reactions that result in the formation of fibrin from fibrinogen, producing the protein network that traps platelets and red blood cells to form a clot. These factors regulate the common coagulation pathway shared by both the intrinsic pathway, activated by exposure of extracellular matrix collagens, and the extrinsic pathway, activated by release of tissue factor by damaged cells. Additionally, some autoantibodies target platelet functions. Platelet activation involves the formation of molecular complexes involving phospholipids, their binding proteins such as GP1b or β2GPI, and VWF and collagens and results in the release of PF4, which neutralizes the natural anticoagulant heparin that is present on most tissues. Platelets can form a temporary clot independent of fibrinogen. Autoantibodies may impair or stimulate platelet function depending on the set of antigens they target.

One possible trigger of autoimmune coagulopathies may be molecular mimicry between SARS-CoV-2 and human blood proteins, which can either result in inappropriate activation of clotting (if the mimic can substitute for the host protein) or induce autoantibodies that inactivate the host protein. Kanduc [39] and Root-Bernstein [40] (Figure 1) have reported extensive mimicry between SARS-CoV-2 proteins and human blood coagulation-related proteins. While Kanduc reported a wider range of possible similarities than did Root-Bernstein, both studies agree that targets of autoantibodies resulting from antigenic mimicry are likely to target collagen 1, GP1b, β2GPI, Factor VIII, Factor IX, PF4, prothrombin (Factor 2), and vWF.

Notably, however, Kanduc and Root-Bernstein both report significant mimicry between a variety of bacterial pathogens and both the SARS-CoV-2 and human blood proteins. Some bacterial proteins mimic SARS-CoV-2 antigens as well, leading Kanduc [39] to propose that prior immunological exposure to these bacterial blood protein mimics might rapidly trigger anamnestic secondary cross-reactive responses of extreme affinity and avidity in the presence of similar SARS-CoV-2 antigens, resulting in thromboembolic adverse events. Root-Bernstein [40], in contrast, has proposed that the SARS-CoV-2 and adenovirus mimicry of human blood proteins is supplemented by complementary bystander bacterial co-infections that result in a hyper-inflammatory state provoking autoimmune disease [40].

Root-Bernstein [40] has suggested that adenovirus vectors or infections may also play a role in stimulating autoimmune coagulopathies following SARS-CoV-2 infection of vaccination since it exhibits an unusual number of antigenic similarities to human blood proteins compared with other viruses (Figure 1) and has been associated in previous studies with coagulopathies (see Discussion below).

However, SARS-CoV-2 alone was unlikely to be a trigger for coagulopathies [39,40,41]. One outstanding reason for this conclusion is that coagulopathies are extremely rare among SARS-CoV-2 vaccinees and those with mild or symptom-free SARS-CoV-2 infections, whereas the incidence of coagulopathies is much higher among moderate and severe COVID-19 cases who are characterized by having an unusually high incidence of viral, fungal and bacterial co-infections (reviewed in [40,41,42,43]). Additional similarity searches were performed by both Root-Bernstein [40] and Kanduc [39] on some of the most common COVID-19-associated bacterial infections (*group A streptococci (GAS), Staphylococcus aureus, Klebsiella pneumoniae, Mycobacterium tuberculosis, Escherichia coli, Haemophilus influenzae* and pathogenic *Clostridia (C. clostridioforme, C. perfringens, C. tetani,* etc.), etc., which revealed that some of the most common autoantigen targets in COVID-19-associated coagulopathies, such as CL, PF4 and β2GP, have significant mimics with these bacteria but do not have significant mimics with SARS-CoV-2 and especially its spike protein [40] (summarized in Figure 1). Since some bacterial infections, especially *GAS*, *Staphylococci, Klebsiella* and *Clostridia*, are themselves associated with increased risks of coagulopathies [44,45,46,47,48,49,50,51,52,53,54,55,56,57,58,59,60,61], these results suggested that preceding or concomitant bacterial co-infections may support the induction of a variety of COVID-19 autoimmune coagulopathies through anamnestic secondary cross-reactivity or bystander activation of complementary autoimmune mechanisms to those activated by SARS-CoV-2. Conversely, if a pre- or co-existing bacterial infection were necessary to induce autoimmune coagulopathies, then the absence of such infections among the vast majority of vaccinees might explain the extremely infrequent occurrence of vaccine-induced thrombotic events [62,63].

The purpose of the research reported here was to explore whether polyclonal antibodies against SARS-CoV-2 proteins, adenoviruses, control viruses, and bacteria associated with moderate-to-severe COVID-19, recognize human blood proteins with sufficient specificity and affinity to be involved in the induction of active autoimmune disease. We also explored whether any interactions occur between the viral and bacterial antibodies and whether any of the antibodies recognize similar SARS-CoV-2 proteins, as predicted by Kanduc [39].

## 2. Results

Results of the ELISA experiments demonstrate significant binding of some SARS-CoV-2, adenovirus, and bacterial antibodies to human blood proteins and the DA-ELISA experiments demonstrate significant binding of some viral antibodies to bacterial antibodies indicating that these antibodies react to antigenically complementary epitopes.

### 2.1. SARS-CoV-2, Adenovirus and Control Virus Antibody Binding to Blood Proteins

The results of virus antibody binding to blood proteins is summarized in Figure 2, statistical analysis is provided in Appendix A, and representative data are provided in Figure 3, Figure 4 and Figure 5. In most cases, the starting concentration of protein used in the experiments was 1 or 10 µM and dilutions were made from there but since cardiolipin and serum albumin are present at much higher concentrations than this in blood serum, 100 µM and 1 mM starting concentrations were used. No binding of any virus antibody to serum albumin or cardiolipin was observed but at least one virus antibody recognized at least one other blood protein. Very weak binding was observed between the SARS-CoV-2 S1 region and PF4 and Factor VIII but given the concentrations of these proteins in blood serum (see below), these results are unlikely to be significant. Significant binding was observed between the spike S1 antibody and vWF, PDE II and collagen 1 (Figure 3 and Figure 4). Notably, the SARS-CoV-2 nucleoprotein antibody also exhibited significant binding to Factor II and collagen 1, and very weakly to Facto IX, suggesting that it may confer additional risks for inducing coagulopathies beyond those inherent in the mimicry of the spike protein for blood proteins. While none of the SARS-CoV-2 antibodies tested bound to β2GP1, adenovirus antibody displayed strong affinity for the protein (Figure 5). Adenovirus antibodies also recognized PF4, vWF, PDE and collagen 1 making the virus potentially a risk for inducing autoimmune coagulopathies with, or without, SARS-CoV-2.

### 2.2. Determining Significance of Antibody Binding Constants

The significance of binding curves summarized in Figure 2 was determined with relation to the median concentrations of the blood proteins in normal human sera, which are summarized in Table 2. Since the Kd represents the concentration at which the antibodies would bind half of the available protein, significant binding was set at any Kd smaller than the median plasma concentration of the protein. In practice, reference to the Kd values demonstrates that the vast majority of binding constants were either significantly lower than median plasma concentration of the protein, indicating that a majority of the protein might be expected to be bound to antibody, or the Kd were significantly greater the median plasma concentration of the protein, indicating that no significant binding would occur. Two proteins are exceptions. One is Platelet Factor 4, which has a very low serum concentration compared with the other proteins but which is present at 400 nM within platelets (see Table 2). We interpret these data to mean that antibody binding to PF4 in vivo is likely to be directed at platelets or at PF4 released during platelet activation rather than at the normally very low amount of PF4 that circulates. Various forms of PDE likewise circulate both in blood serum and it is stored within platelets making both potential targets for autoimmune reactions. Notably, the binding constants of antibodies tested here that recognized human blood proteins with significant affinities are in the same nanomolar range as are polyclonal antibodies against their own antigens (10^−7^ to 10^−9^ M) [64] and are therefore of potential clinical significance. In every case, the binding constants of the significant interactions reported here are smaller than the concentrations of the antigens in sera or cells so that antibody binding to these antigens is highly favored.

### 2.3. Bacterial Antibody Binding to Blood Proteins

The results of bacterial antibody binding to blood proteins is summarized in Figure 6, statistical analysis is provided in Appendix A, and representative data are provided in Figure 7, Figure 8 and Figure 9. The same concentrations of antibodies and proteins were used in these experiments as in the virus antibody experiments in Section 2.1 and significance of the resulting binding constants was evaluated as in Section 2.2. Although most of the bacteria demonstrated significant sequence similarities to human serum albumin in a previous BLAST study [40], none of the antibodies tested cross-reacted with serum albumin even at 1–4 mM concentrations. Most of the antibodies did, however, recognize cardiolipin as an antigen, and often at least one other blood protein, as previously predicted [41]. Significant binding to almost all of the blood proteins was observed with group A streptococcal antibodies making it a particularly important candidate for stimulating COVID-19 coagulopathies if present as a co-infection with SARS-CoV-2. *Clostridium* and *M. tuberculosis* are, by the same reasoning, the least likely to initiate or participate in stimulating COVID-19-related coagulopathies.

### 2.4. DA-ELISA Results: Virus Polyclonal Antibody Binding to Bacterial Polyclonal Antibodies

Many of the proteins involved in this study are known to bind to each other, and are therefore molecularly complementary to each other, as part of the blood coagulation pathways, providing the possibility for the expression of complementary antigenic regions. Because many of these binding pairs segregate to some extent between the bacteria and the viruses tested in Section 2.1 and Section 2.3 above, it is therefore possible that the bacterial antibodies will be complementary to the virus antibodies so that the protein–protein binding is reflected in antibody-antibody binding.

Focusing specifically on the proteins tested here, well-established binding interactions that relate to the proteins tested here are summarized in Figure 10. When collagen 1 is revealed during tissue injury, VWF, GP1b, glycoprotein VI, as well as clotting factors, can bind to it initiating platelet aggregation and hemostasis [75,76]. VWF binds to GP1b and Factor IX In the common coagulation pathway [28,77,78], activated Factor VIII binds to Factor IX activating it; activated Factor IX binds to Factor X activating it; activated Factor X then binds to Factor 2 (prothrombin) converting it into thrombin [77,78,79]. VWF also has binding sites for Factor VIII and β2GPI [76,77,78] ensuring localization of these diverse reactions. β2GPI also binds to PF4, F2, Factor X and CL [80]. In addition, platelet and endothelium phosphodiesterases bind CL [81,82], as do complement C1q-binding proteins, suggesting that cardiolipin and C1q share common epitopes, and providing a mechanism for activation of the complement cascade by anti-CL antibodies [83].

Additionally, adenoviruses, including ChAdOx1, the vector for the ChAd Ox1 nCoV-19 vaccine bind directly to PF4, demonstrating the presence of complementary antigens on the virus [84]. *Streptococci* (e.g., *pyogenes* and *pneumoniae*) and *Staphylococci* (e.g., *aureus*) are also able to initiate clot formation through contact initiation involving binding of their surface proteins to vWF and Factor XIII [79,85]. This process appears to be initiated by collagen-like regions expressed by streptococcal collagen-like (Scl) proteins in pathogenic *Streptococci* [86] and by von Willebrand Factor binding protein-like regions on *Staphylococci* [87]. It follows that if SARS-CoV-2 (or any of the other viruses studied here) have proteins that mimic vWF or PF4, as the previous Results suggest, they, too might be able to bind to *Streptococci* and or *Staphylococci*. Antibodies against such complementary epitopes might also result in complementary antibody idiotypes that behave like idiotype-anti-idiotype pairs.

As a result of these many and diverse types of protein–protein and blood protein-microbe interactions, virus-induced and bacteria-induced antibodies may each mimic the binding characteristics of one or more blood proteins and their complementary interactions with each other. The consequence of such complementary interactions would be for some SARS-CoV-2 or adenovirus antibodies to recognize some bacterial antibodies as antigenic targets. This possibility was explored using DA-ELISA to test for antibody-antibody binding. The results of these experiments are summarized in Figure 11, statistical analysis is provided in Appendix A, and representative data are provided in Figure 12 and Figure 13. Significant binding was observed between antibodies elicited by at least one portion of the SARS-CoV-2 spike protein and antibodies against *Streptococci*, *Staphylococcus aureus* and *Klebsiella pneumoniae*. Since all of these antibodies also recognized one or more blood proteins as antigens, these SARS-CoV-2-bacterial antibody interactions are likely to be of significance for understanding blood coagulopathies in COVID-19. Additionally, coxsackievirus type B antibodies recognize *Streptococcal* and *Staphylococcal* antibodies as antigenic targets, as well as *E. coli* and *Clostridium* antibodies, and HSV1 antibodies recognized *Staphylococcal* antibodies. These latter interactions are unlikely to shed light on COVID-19 coagulopathies since these viral antibodies failed to recognize any blood protein tested here. Their possible role in other autoimmune diseases will, however, be explored in the Discussion below.

### 2.5. Results Summary

In sum, SARS-CoV-2 and adenovirus antibodies each recognized multiple human blood proteins as antigens, particularly vWF, PDE II and collagen 1. The adenovirus antibodies were also notable for having significantly more affinity for PF4 (a major target of autoimmune coagulopathies in COVID-19) than SARS-CoV-2 antibodies. Antibodies against bacteria that are commonly found as co-infections in moderate-to-severe COVID-19, particularly *Streptococci*, *Staphylococci*, *Klebsiella*, and *E. coli*, also recognized multiple blood proteins, most notably CL, β2GPI, and PF4. These bacterial antibody targets are molecularly complementary to the SARS-CoV-2 antibody targets. This complementarity was reflected in the DA-ELISA experimental results demonstrating that *Streptococci*, *Staphylococci*, and *Klebsiella* antibodies recognized SARS-CoV-2 antibodies as antigens.

## 3. Discussion

The results of this study are summarized in Figure 14, which displays with X’s the statistically significant rates of increased similarities between SARS-CoV-2, its spike protein, adenoviruses, other virus controls and bacteria associated with moderate-to-severe COVID-19 and compares them with significant binding to human blood proteins by antibodies against these viruses and bacteria, represented with blue boxes.

As noted in the Introduction, the studies by Kanduc [39] and Root-Bernstein [40] agreed that among the proteins that are known targets of autoantibodies in SARS-CoV-2-associated coagulopathies, GP1b, β2GPI, Factor VIII, Factor IX, PF4, prothrombin (Factor 2), and vWF exhibit high degrees of similarity with SARS-CoV-2. However, of these only Factor IX, PF4 and vWF were experimentally found to be recognized by SARS-CoV-2 antibodies. PDE (which was on Root-Bernstein’s list) and collagen 1 (which was on Kanduc’s list) were also recognized by SARS-CoV-2 antibodies. Notably, the SARS-CoV-2 spike protein, which has been the focus of most vaccine development, displayed significant cross-reactivity limited to only three of the ten blood proteins tested: PDE II, vWF and collagen 1 (Figure 2). This limited cross-reactivity may explain why the vast majority of people vaccinated against SARS-CoV-2 using spike-protein-based agents fail to develop coagulopathies. Additional factors, to be discussed below, may be required to complement this limited cross-reactivity to boost it to active autoimmune disease.

Root-Bernstein [40] also reported significant similarities between blood proteins and proteins of bacteria associated with SARS-CoV-2 co-infections. The Results confirmed that some of these bacterial-blood protein similarities result in the production of antibodies cross-reactive with some of these blood proteins. *Staphylococci*, *Streptococci*, *Klebsiella* and *E. coli* were the most important of these in terms of number of proteins recognized by their antibodies (Figure 6). Notably, the blood proteins recognized by bacterial antibodies tend to be a different group than those recognized by SARS-CoV-2 or adenovirus antibodies in accordance with the predicted sequence similarities summarized in the Introduction (Figure 1). The bacterial antibodies tended to recognize CL, β2GPI and PF4 rather than vWF, PDE and collagen 1, which are the main targets of SARS-CoV-2. The difference in blood protein targets between bacteria and SARS-CoV-2 is likely an important clue as to the pathogenesis of COVID-19 coagulopathies that will be taken up in detail below. For the moment, it is interesting to note that Kanduc [39] appears to be correct in predicting that some bacteria share common epitopes with SARS-CoV-2 resulting in similar antibody affinities: *Streptococci*, for example, and SARS-CoV-2 antibodies each recognized PF4, F2, and vWF (Figure 14).

The utility of similarity searches for predicting experimental results can be evaluated from the Results reported here. Root-Bernstein [40] accurately predicted 19 microbial interactions with human proteins and 72 instances where no binding was observed, while missing 11 microbial interactions with the proteins that were experimentally verified and making 18 predictions that were falsified. So, overall, of 120 possibilities, 91 predictions were verified (76%) and 29 (24%) were not. Most of the inaccurate predictions were among the bacteria, serum albumin making up a major group for which no cross-reactivities were observed but many predicted. The poorer results for the bacterial similarities is likely the fault of the search method employed, which was a simplistic BLAST procedure exploring using each blood protein to search for similarities in the entirety of the bacterial database. The results for the viruses, in contrast, involved a more sophisticated procedure that searched each blood protein against one viral protein at a time using LALIGN for pairwise comparisons. The virus searches yielded much more accurate results, with 54 of the 60 predictions validated (90%) and only six of the 54 predictions being inaccurate (10%). No predictions were made regarding virus antibodies binding to bacterial antibodies as no theoretical basis currently exists to make such predictions.

Our results can also be compared with those from previous studies that have used both rodent and human antibody preparations. Vojdani and Kharrazian [88] have previously explored binding of both SARS-CoV-2 mouse and rabbit monoclonal antibodies to a variety of human proteins, a handful of which overlapped the proteins studied here. They identified significant binding of spike protein and nucleoprotein antibodies to collagen (type undefined), generally confirming the results reported here (Figure 2). Their subsequent study of human SARS-CoV-2 monoclonal antibody targets [89] verified the mouse and rabbit results by demonstrating weak collagen binding for spike, nucleoprotein and membrane protein antibodies but not envelope protein antibodies. They did not find significant binding to platelet glycoproteins in either study, which is not consistent with our observations but it must be emphasized that, as opposed to the quantitative ELISA method utilized here, both Vojdani and Kharrazian studies used monoclonal antibodies rather than polyclonal ones and binding was studies at one concentration of antibodies to beads or wells coated with one concentration of protein so that the effect of in vivo protein concentration on antibody binding could not be determined.

Our rodent antibody studies are consistent with and confirm Greinacher et al.’s [20] previous report that cross-reactivity between SARS-CoV-2 spike protein polyclonal antibodies isolated from human patients and platelet factor 4 is probably not of sufficient affinity to contribute to disease (Figure 2). We were not, however, able to confirm Passariello et al.’s report [90] that SARS-CoV-2 spike RBD protein antibodies cross-react with PF4, although we did observe weak cross-reactivity between antibodies against SARS-CoV-2 matrix protein and PF4. The affinities are such that spike and matrix antibodies would not be able to recognize the concentration of PF4 normally free in solution (Table 2) but might be effective at the concentrations of PF4 released upon platelet activation. Overall, the evidence suggesting that SARS-CoV-2 spike protein vaccination may contribute to PF4 autoantibody production is poor and contradictory. Because PF4 is a major target of autoantibodies in COVID-19 vaccine-associated coagulopathies, lack of clear cross-reactivity of SARS-CoV-2 antibodies to PF4 again helps to explain the safety of the spike-protein-based vaccines. People who do develop anti-PF4 antibodies following SARS-CoV-2 vaccination may need to be exposed to other microbes, such as adenoviruses or bacteria, that can induce more robust PF4 cross-reactivity (see Figure 2, Figure 6 and Figure 14).

As was just noted, these observations concerning the non-cross-reactivity of the spike protein with PF4 strongly suggest that some other source of PF4 antibody activation is required, which we have identified as either a bacterial infection (*Streptococci*, *Staphylococci* and *E. coli* being the most likely) (Figure 6), or, in the case of adenovirus-vectored SARS-CoV-2 spike protein vaccines adenovirus [91] (Figure 2). Notably, as mentioned in the Introduction, transient anti-PF4 antibodies have been demonstrated in 30 to 75% of adenovirus-vectored COVID-19 vaccinees, though autoimmune coagulopathies have remained extremely rare (24, 27, 29). Another source of PF4 autoantibody production may be natural adenovirus infections, which complicate 5 to 7% of hospitalized SARS-CoV-2 patients [92,93,94] and are present in a very high percentage of hepatitis A-infected patients [95]. These adenovirus co-infections may activate PF4 autoantibodies by means of the observed cross-reactivity between viral antibodies and PF4 reported here. Notably, SARS-CoV-2 complicated with adenovirus co-infection has a significantly increased probability of hospitalization and death compared to an uncomplicated SARS-CoV-2 infection but not as a result of increased need for ventilation, indicating that cardiovascular complications are a more likely cause of enhance morbidity and mortality [93]. Influenza A, herpes simplex type 1 and coxsackievirus antibodies had no cross-reactivities with human blood proteins, highlighting the unusual risks of autoimmune coagulopathies posed by SARS-CoV-2 and adenoviruses both individually and in combination.

Another notable result reported here is that none of the antibodies against any SARS-CoV-2 protein cross-reacted with CL which, like PF4, is a target of autoantibodies in most COVID-19 coagulopathies. Anti-CL IgG and IgM are rarely (4.5–5.7 and 6.4–6.6%., respectively) observed in COVID-19 hospitalized patients who are not diagnosed with thrombosis [26,34] but anti-CL IgG and IgM antibodies are present in 52% and 40%, respectively, of COVID-19 patients admitted to intensive care and are highly associated with thrombotic events [96]. CL (3-bis(sn-3’-phosphatidyl)-sn-glycerol) is a phospholipid that is found as a normal component of blood plasma as well as the mitochondrial membrane. It is not found in any known virus but is common to the cell membranes of all the bacterial species studied here [40]. Thus, a number of studies have recently reported that vaccination with the SARS-CoV-2 vaccines (adjuvanted and adenovirus vectored) fails to induce anti-CL antibodies [96,97,98,99]. These studies using human sera and polyclonal antibody preparations are consistent with our observation of the lack of cross-reactivity between SARS-CoV-2 or adenovirus rodent antibodies and CL. This raises the important question of what does induce anti-CL antibodies in COVID-19-associated coagulopathies and the answer, as was also the case with anti-PF4 antibodies, is most likely bacteria. As noted above, all bacteria that occur as co-infections with SARS-CoV-2 express CL in their membranes and most of the anti-bacterial antibodies tested in this study cross-reacted with CL (Figure 14).

Notably, COVID-19 patients characterized by anti-CL antibodies are also characterized by having anti-β2-glycoprotein I (aβ2-GPI) IgG and IgM (39% and up to 34% of patients, respectively) and up to 12% are also positive for lupus anticoagulant (LA) antibodies [96], another recognized risk factor for coagulopathies. The confluence of these sets of antibodies strongly argues for co-infections of SARS-CoV-2 with bacteria. Even more importantly, some clinical evidence suggests that such co-infections may be necessary to induce autoimmune coagulopathies since even in ICU COVID-19 cases, the presence of only a single anti-phospholipid (aPL) antibody was not correlated with development of coagulopathies [26,34,100,101] Patients at risk for coagulopathies almost invariably displayed the presence of two or more of the autoantibodies already mentioned: PF4, CL, aβ2-GPI and/or LA.

The correlation between coagulopathies and the presence of multiple anti-phospholipid antibodies in a single patient, but lack of correlation to individual autoantibodies, suggests that coagulopathies result from the complementary interaction of several autoantibodies rather than from the presence of a single autoantibody type. This conjecture is given further credence by the fact that CL or oxidized CL in lipoproteins complexes with plasma proteins such as β2-GPI I, prothrombin, protein C, or protein S and with platelet or endothelial surface proteins such as PF4, PDE and collagens [102]. Once again, it is notable that the presence of both anti-CL and anti-β2-glycoprotein I together correlated with risk of thrombosis [103]. Thus, a pre-COVID-19 study of heparin-induced thrombocytopenia (HIT) found that among 30 patients with lupus anticoagulant, 25 also displayed anti-CL antibodies, 21 anti- aβ2-GPI, and 18 anti-prothrombin (F2) antibodies [104]. Indeed, the epitopes for some antiphospholipid antibodies are adducts of oxidized phospholipid and β2-glycoprotein I and these other proteins [105], an observation that formally demonstrates the antigenic complementarity of CL for these proteins.

In light of the results reported here, the evidence that people who develop COVID-19 related coagulopathies are at highest risk for bacterial and viral co-infections, and that such people are characterized by developing multiple autoantibodies against complementary blood proteins, the following model is proposed for the pathogenesis of these coagulopathies.

To begin with, we have demonstrated here that some bacterial and viral epitopes mimic human blood protein epitopes so that antibodies against the microbes cross-react with these blood proteins. Because many of the blood proteins are molecularly (and therefore antigenically) complementary to each other (Figure 15), some of the antibodies that cross-react with these blood proteins will form circulating immune complexes (CIC) comprised of antibodies complementary to each other, and probably incorporating the microbial and blood proteins as well (Figure 16). Some of these CIC will be composed of antibodies against SARS-CoV-2 and/or adenovirus in combination with antibodies against bacteria that may be present in the same patient. We have demonstrated experimentally (Figure 11, Figure 12 and Figure 13) that such complexes do form between SARS-CoV-2 or adenovirus antibodies and antibodies against *Streptococci, Staphylococci* and *Klebsiella pneumoniae* and we interpret these complexes as experimental equivalents of the circulating immune complexes (CIC) that occur in moderate-to-severe cases of COVID-19. Circulating immune complexes can directly activate platelets (Figure 17) by initiating complement binding [106,107,108], have been observed as a cause of platelet activation in COVID-19-related contexts [109,110,111], often contain PF4-reactive IgG [112,113,114] (and thus bacteria-induced antibodies), and participate in the formation of NETosis [115,116]. CIC formation therefore provides one mechanism by which combined SARS-CoV-2-bacterial co-infections can induce coagulopathies.

A concomitant or alternative set of reactions affecting coagulation may also be initiated by combinations of SARS-CoV-2, adenovirus and bacterial antibodies. Some of the antibodies elicited by the viruses and bacteria cross-react with antigens on blood system cells such as platelets, red blood cells (RBC), the vascular system, and/or the soluble proteins in blood serum involved in the coagulation pathway (Figure 18). The cross-reactivities of viral and bacterial antibodies with blood coagulation proteins demonstrated in the Results here and summarized in Figure 14 may result in some of these antibodies targeting platelets, RBC and the vasculature causing cellular or tissue damage. When such damage reveals collagens (a target of some of the potential autoantibodies described in our Results), this initiates platelet activation: the collagen binds vWF in concert with GP1b and platelet glycoprotein VI (Figure 19). Other cellular targets of cross-reacting bacterial and viral antibodies that result in cellular or tissue damage may include CL and PDE. Activation of platelets by binding to collage results in release of PF4, b2GPI and stimulates the common coagulation pathway involving some of the coagulation factors (e.g., F2, FVIII, FIX and FX) described in the Introduction and Results. Only some of these are shown in Figure 20 for the sake of simplicity. Depending on the particular set of autoantibody reactions that are initiated by any particular SARS-CoV-2-bacteria combination (with the possible additional participation of adenovirus), stimulation and or interference with blood coagulation at multiple points in the pathway is possible.

A concomitant or alternative set or reactions affecting coagulation may also be initi-ated by combinations of SARS-CoV-2, adenovirus and bacterial antibodies. Some of the antibodies elicited by the viruses and bacteria cross-react with antigens on blood system cells such as platelets, red blood cells (RBC), the vascular system, and/or the soluble pro-teins in blood serum involved in the coagulation pathway (Figure 18, Figure 19 and Figure 20).

A key point to be drawn from the model just presented is that because different co-infections exhibit different sets of blood protein cross-reactivities, different co-infections are likely to trigger different types of coagulopathies, therefore explaining the range of different coagulation complications that can beset COVID-19 patients. The high rate of bacterial and viral co-infections among severe COVID-19 patients explains their concomitantly high rate of coagulopathies, while the very low rate of such co-infections among mild COVID-19 patients and among SARS-CoV-2 vaccinees explains their correspondingly very low rates of coagulopathies.

Various implications follow from the model just proposed. One is that vaccination against pneumococci such as GAS should lower the risk of severe COVID-19. Evidence from several studies using a variety of methods suggests that this is the case [117,118,119,120,121,122,123,124,125]. A further implication that has not been tested directly is that pneumococcal vaccination should specifically lower the risk of COVID-19-associated coagulopathies. Additionally, it has been observed that rates of pneumococcal infections have dropped significantly since the adoption of personal protective measures such as masking and distancing [126,127]. While the drop in rates of hospitalizations and deaths as a proportion of SARS-CoV-2 infections is usually attributed to selection for less pathogenic SARS-CoV-2 strains and increasing rates of SARS-CoV-2 vaccination, the model just presented suggests that lowering the risk of bacterial co-infections by means of pneumococcal and perhaps *Haemophilus influenzae* vaccination may also be protective and have the effect of significantly reducing severe COVID-19 complications such as coagulopathies and myocardial complications as well.

One important factor that is not directly addressed by the model, but is present indirectly, is that autoimmune diseases generally require support from the innate immune system in the form of hyperinflammation mediated by increased release of cytokines [42,128]. COVID-19 patients who develop evidence of cytokine overproduction syndromes (a “cytokine storm”) are among those who are most at risk for coagulopathies, and Root-Bernstein has demonstrated that such cytokine overproduction syndromes are usually, if not always, associated with the presence of polymicrobial infections [42,128]. It is known, for example, that SARS-CoV-2 antigens activate TLRs 3, 7 and 9 whereas bacterial glycoproteins activate TLRs 1 and 2 and CL (a lipopolysaccharide found only in the bacteria in this study) activates TLR4; TLRs 2 and 4, in turn, synergize with TLRs 3 and 9 to produce a hyperinflammatory state (reviewed in [42,128]). Thus, a substantial body of evidence beyond the current study also points towards COVID-19 coagulopathies being due to bacterial, and possibly adenoviral, infections complicating SARS-CoV-2 and activating synergistic sets of Toll-like receptors resulting in a hyperinflammation. This effect has been demonstrated for a combination of SARS-CoV-2 with *Streptococci* in mouse models of coinfection [129,130]. Notably, this hyperinflammatory state could be prevented by vaccination against either SARS-CoV-2 or pneumococci.

Further tests of the results reported here are clearly possible and may validate or invalidate the significance of our data. In the first place, only some of the blood proteins that may be involved in coagulopathies have been explored here so that a number of other blood proteins targets such as complement proteins, platelet glycoprotein 1b, Rhesus blood factors, etc., need to be tested for cross-reactivity with SARS-CoV-2, adenovirus and bacterial antibodies. Additionally, antibodies against a number of bacterial coinfections that are often found in severe cases of COVID-19, such as *Acinetobacter baumannii*, *Haemophilus influenzae*, *Pseudomonas aeruginosa*, *Mycoplasma pneumoniae*, etc. [131,132,133,134] remain to be explored for cross-reactivities with blood proteins implicated as autoimmune targets in COVID-19 coagulopathies.

CIC produced artificially, in vitro, by combining SARS-CoV-2 antibodies with complementary bacterial antibodies could be tested to determine whether they activated platelets. More importantly, the CIC naturally occurring during COVID-19-associated coagulopathies be investigated to determine whether they contain both SARS-CoV-2-induced antibodies as well as bacterial- or adenovirus-induced antibodies, as predicted here.

Other implications of our data can be tested in animal models. For example, SARS-CoV-2-susceptible species (such as golden hamsters or some strains of mice) might be co-infected with the virus and with bacteria such as group *A Streptococci, Staphylococci, Klebsiella, Acinetobacter* or *Haemophilus*, or with viruses such as adenovirus type 5 (e.g., [129,130]). The effect of such bacterial coinfections on vaccination with SARS-CoV-2 vaccines could be tested in a similar manner. Our prediction is that animals infected with only SARS-CoV-2 or its vaccine or only with the bacterium alone will not develop coagulopathies (although some are likely, as in human patients, to develop non-pathogenic autoantibodies), while coinfected animals will demonstrate increased rates of thrombocytopenia and/or thromboses. Alternatively, since it is assumed here that COVID-19 coagulopathies are autoimmune diseases, it should be possible to inoculate naïve rabbits with combinations of polyclonal (rabbit) antibodies against the SARS-CoV-2 proteins (e.g., spike, nucleoprotein or whole virus) in combination with (rabbit) polyclonal antibodies against group A *Streptococci*, *Staphylococci*, etc. Such combinations are predicted to produce clinically evident coagulopathies. Correspondingly, we predict that rabbits inoculated with only the SARS-CoV-2 antibodies or only with the bacterial antibodies will not develop coagulopathies.

## 4. Materials and Methods

### 4.1. Enzyme-Linked Immunosorbent Assay

(ELISA) was used to investigate cross-reactivities between microbial antibodies and blood coagulation-related proteins. The tissue protein was diluted in pH 7.0 phosphate buffer to a concentration of 1 μM. This standard solution was then diluted by three-fold steps with the buffer to about 10–11 M. Two wells received only phosphate buffer as controls. All samples were run in duplicate. A total of 100 μL of each protein dilution was added to wells of a Costar round-bottomed 96-well ELISA plate and incubated for one hour. The excess protein was triply washed out using a 1% Tween 20 solution in phosphate buffer and a plate washer. Next, 200 μL of blocking agent (2% polyvinylalcohol in phosphate buffer) was added to every well, incubated for an hour, and then triply washed. (Bovine serum albumin, which is more commonly used as a blocking agent, was not used in this study because of the possibility that SARS-CoV-2 antibodies might cross-react with serum albumin—see Figure 1). An antibody against a microbe (at 1 mg/mL concentration) was then diluted to 1/500 in phosphate buffer and 100 μL added to every well. The antibody was incubated for an hour and then triply washed. A species-appropriate horse-radish peroxidase-linked secondary antibody was then at a dilution of 1/1000, incubated for an hour, and triply washed. Finally, 100 μL of ABTS reagent (Chemicon, Temecula, CA, USA) was added, incubated for 15 min, and the plate read at 405 nm in a Spectramax UV-VIS scanning spectrophotometer (Molecular Devices, LLC., Palo Alto, CA, USA). Data were gathered using Spectramax software and then analyzed using Excel. Analysis essentially consisted of subtracting non-specific binding to the buffer-only wells from the protein-containing wells and plotting the amount of antibody binding (as measured by absorbance at 405 nm) as a function of protein concentration. Binding constants (Kd) were calculated by finding the inflection point of the resulting curve using a curve-fitting program (https://mycurvefit.com/ (accessed on 7 July 2022)) that also provided R2 and SE values. All experiments were run in duplicate and some were replicated more than once either to confirm results that are crucial to the interpretation of the results or in the case of ambiguous data.

### 4.2. Double Antibody ELISA

(DA-ELISA) was used to investigate possible antigenic complementarity between the antibodies used in the study. DA-ELISA differs from ELISA in that the protein laid down in n the 96-well plate in the initial step of an ELISA is substituted with an antibody. A second antibody (from a different species) is tested for its ability to bind to the first. The ability of the second antibody to bind to the first is then monitored using peroxidase-linked antibody against the species from which the second antibody is derived [135,136,137,138]. As in the ELISA protocol, the first antibody is made up at a concentration of about 12 μM (assuming IgG antibodies have a molecular weight of 150,000 daltons) and then serially diluted by factors of three. The rest of the protocol is the same as for the ELISA.

In general, it is only possible to run DA-ELISA with antibodies from species sufficiently genetically different that the peroxidase-linked antibody against the second species does not recognize the antibodies from the first species. One exception is when one of the antibodies is already conjugated to an enzyme such as horse radish peroxidase (HRP). In the latter case, it is possible to test for HRP-conjugated antibody binding to another antibody (without HRP) of the same species that has previously been adsorbed to the ELISA plate. Sometimes, there is, however, cross-reactivity between supposedly species specific HRP-conjugated antibodies. We found that in this study that anti-rabbit-horse-radish-peroxidase (HRP) antibodies recognized antibodies raised in guinea pigs and vice versa so that combinations of rabbit versus guinea pig DA-ELISA was not possible.

### 4.3. Antibodies

Six rabbit anti-SARS-CoV-2 antibodies from two suppliers were employed, with reactivities against the Spike 1, Spike 2, RBD, Matrix (or membrane), Nucleocapsid and Envelope proteins. Polyclonal antibodies (derived either from goat or rabbit, with two exceptions when antibodies from these species were not available) against adenoviruses, influenza A, human herpes simplex type 1, *Clostridia*, *Escherichia coli*, *Klebsiella pneumoniae*, *Staphylococcus aureus*, and *group A streptococci* were also employed, as well as a blend of monoclonal antibodies against coxsackievirus B types 1–6. These antibodies are listed in Table 3. The Invitrogen and Millipore antibodies against SARS-CoV-2 Spike protein 1 yielded identical results and were used interchangeably in the reported experiments.

### 4.4. Proteins

Table 4 shows the proteins utilized in the experiments, as well as their suppliers. While we attempted to source human proteins, considerations of cost sometimes made this impossible so that either murine or bovine proteins were substituted. In these cases, a homology search was performed using BLAST 2.0 on the Expasy website (https://www.expasy.org (accessed on 3 May 2021)) to ensure that the human protein did not differ by more than one percent (and usually only a couple of amino acids) from its non-human source. A direct test of group A streptococcus antibody binding to both murine and human coagulation factor IX found no significant differences in the binding curves and the curves yielded the same binding constant.

### 4.5. Statistics

Since binding constants were determined by identifying the inflection point of the binding curves, the data used to generate the curves was subjected to two statistical tests, standard error (SE) and the coefficient of determination (R^2^) in order to assess the degree of confidence that can be assigned to the binding constants. SE and R^2^ were calculated, and the inflection point determined, using a basic nonlinear exponential curve fitting equation at MyCurveFit (https://mycurvefit.com/ (accessed on 7 July 2022)).

### 4.6. Methodological Limitations

The main methodological limitation of this study is that the antibodies used are mostly rabbit or goat, since human polyclonal antibodies or sera against the range of viruses and bacteria tested do not appear to be available and using the human monoclonal antibodies against SARS-CoV-2 that are available would not address the possibility of cross-reactivity arising from human polyclonal responses. Furthermore, use of human sera was discouraged by the inability to determine whether the patients donating the sera had previously been vaccinated against or infected with any of the viruses or bacteria other than SARS-CoV-2 that were investigated in this study. We were also unable to identify sources of human polyclonal antibodies against SARS-CoV-2 specifically from patients suffering from coagulopathies. Thus, simplicity and clarity of results mandated use of non-human antibodies; however, extrapolation to human immune responses cannot be assured. An additional limitation of the study was that it does not address possible cross-reactivity of SARS-CoV-2 antibodies with any of the Rhesus factor, ADAMTS, CD55, or Complement, or other blood protein antigens that have been found to display significant similarities to SARS-CoV-2 antigens (Figure 1 and [39,40]); sources for adequate quantities of these proteins were not found at affordable prices. Thus, some key potential autoimmune targets on vascular cells, red blood cells and platelets remain to be explored for SARS-CoV-2 cross reactivity. Furthermore, no attempt to extend the antibody binding studies performed here to cellular or organismal models was attempted.

## Figures and Tables

**Figure 1 ijms-23-11500-f001:**
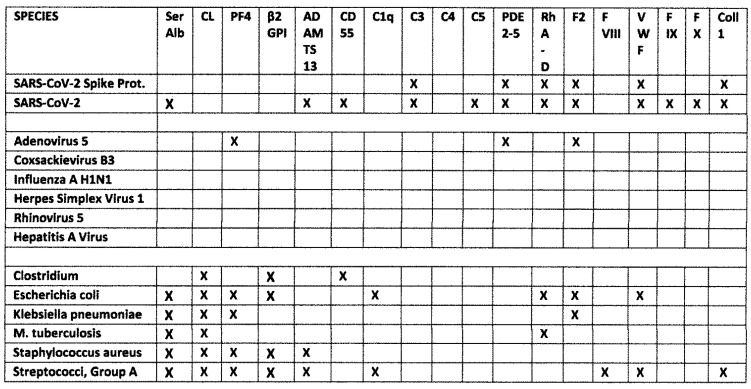
Predictions (indicated by X’s) of cross-reactivity based on significantly increased frequency of human blood protein mimics present in SARS-CoV-2, its spike protein, adenovirus 5, five other viruses (three newly added to this figure), and six bacteria associated with severe COVID-19 [39,40]. The X’s in the virus rows indicate that the number of similarities between a virus protein and a human blood protein was at least three times the average number found among the control viruses. The X’s in the bacteria rows indicate that at least one high-similarity, low-probability sequence was shared with the human blood protein [40]. CL = cardiolipin; Ser Alb = human serum albumin; β2GP = beta 2 glycoprotein; ADAMTS13 = von Willebrand factor-cleaving protease or VWFCP; CD55 = Complement decay-accelerating factor; C1q, C3, C4 and C5 are complement proteins 1q, 3, 4 and 5; PDE = phosphodiesterase; Rh = rhesus factor; F2, FVIII FIX and FX are blood coagulation factors; VWF is von Willebrand’s factor; Coll 1 is collagen type 1.

**Figure 2 ijms-23-11500-f002:**
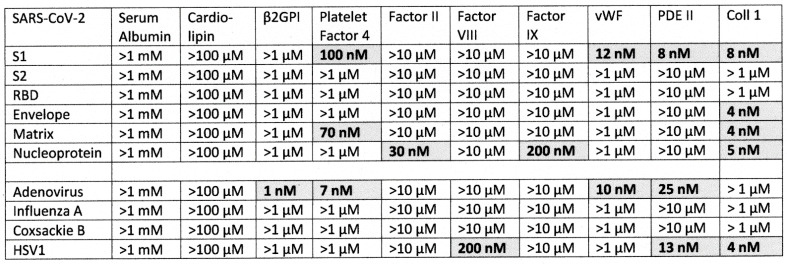
Summary of results of ELISA experiments exploring binding to blood proteins by SARS-CoV-2 polyclonal antibodies (rabbit) against the S1, S2, RBD, envelope, matrix and nucleoproteins, as well as experiments employing polyclonal antibodies against adenoviruses, influenza type A virus (H1N1), coxsackie B virus and human herpes simplex type 1 virus (HSV1). Binding constants were determined from the inflection point of the binding curves, some of which are illustrated in Figure 3, Figure 4 and Figure 5. Significant binding is indicated by bolded numbers against a grey background. As noted in the Methods, S1 antibodies from different vendors were interchangeable, yielding identical results and are not distinguished here. For the definition of significance, see Section 2.2. β2GPI = beta 2 glycoprotein I; vWF = von Willebrand Factor; PDE = phosphodiesterase II; Coll 1 = collagen.

**Figure 3 ijms-23-11500-f003:**
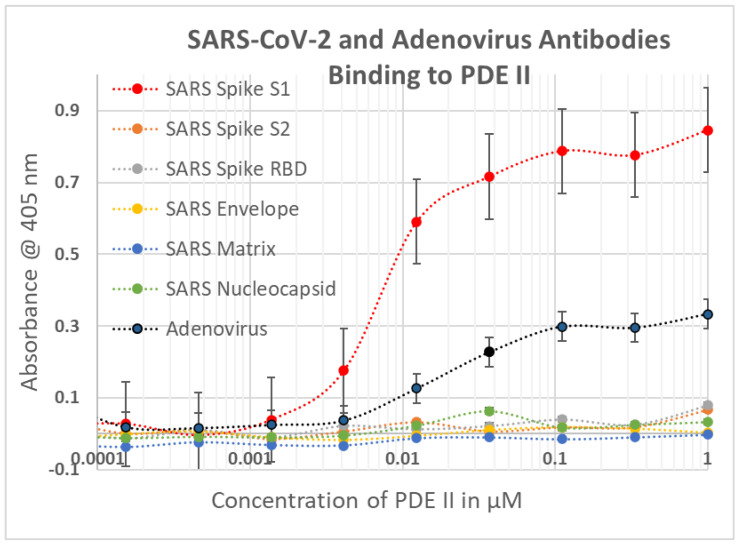
ELISA results showing SARS-CoV-2 (SARS) and adenovirus polyclonal antibodies (rabbit) binding to phosphodiesterase II (PDE II). Experiments were run in duplicate.

**Figure 4 ijms-23-11500-f004:**
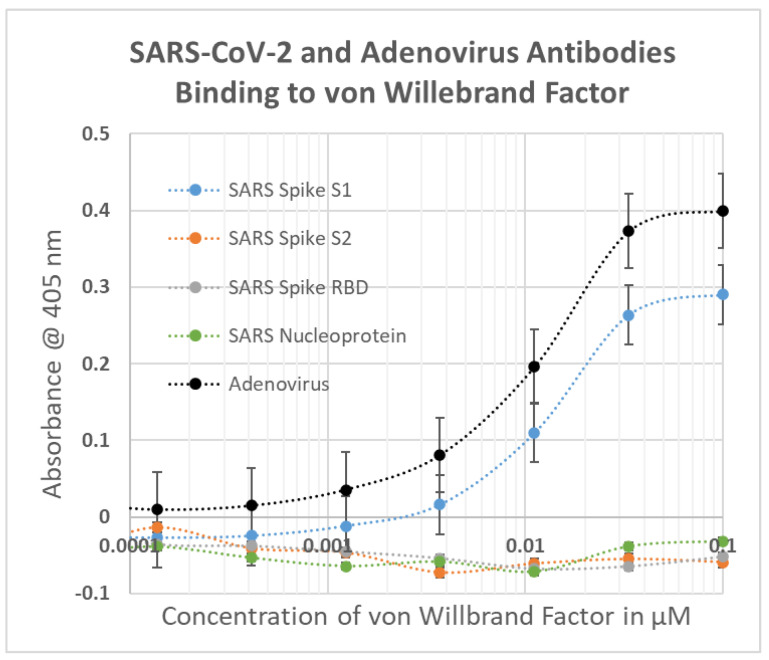
ELISA results showing SARS-CoV-2 (SARS) and adenovirus polyclonal antibodies (rabbit) binding to von Willebrand Factor (vWF). Experiments were run in duplicate.

**Figure 5 ijms-23-11500-f005:**
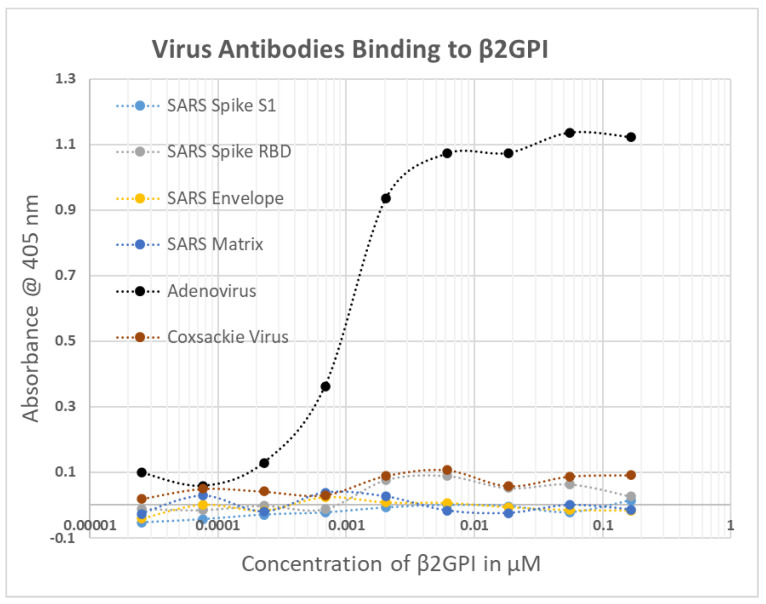
SARS-CoV-2 (spike S1, spike RBD, envelope, matrix), adenovirus and coxsackie Virus (type B) antibody binding to beta 2 glycoprotein I (β2GPI). These are the results of a single experiment, which is why there are no error bars. Results were confirmed in by separate replication.

**Figure 6 ijms-23-11500-f006:**
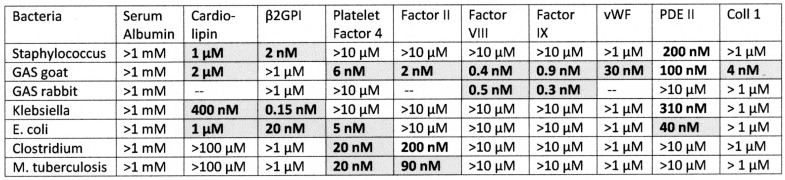
Summary of results of ELISA experiments exploring binding to blood proteins by bacterial polyclonal antibodies (rabbit or goat). Not all proteins were tested with rabbit-anti-GAS antibody: the missing tests are indicated by dashes. Binding constants were determined from the inflection point of the binding curves, some of which are illustrated in Figure 7, Figure 8 and Figure 9. Significant binding is indicated by bolded numbers against a grey background. For the definition of significance, see Section 2.2. GAS = *group A streptococci* antibody; *E. coli* = *Escherichia coli* antibody; *M. tuberculosis* = *Mycobacterium tuberculosis* antibody; β2GPI = beta 2 glycoprotein I; vWF = von Willebrand Factor; PDE = phosphodiesterase II; Coll 1 = collagen 1.

**Figure 7 ijms-23-11500-f007:**
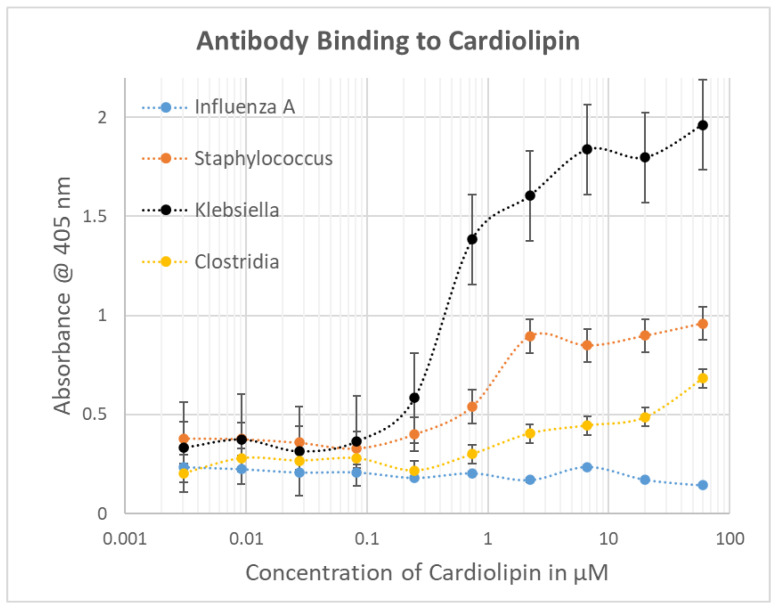
Bacterial antibody binding to cardiolipin.

**Figure 8 ijms-23-11500-f008:**
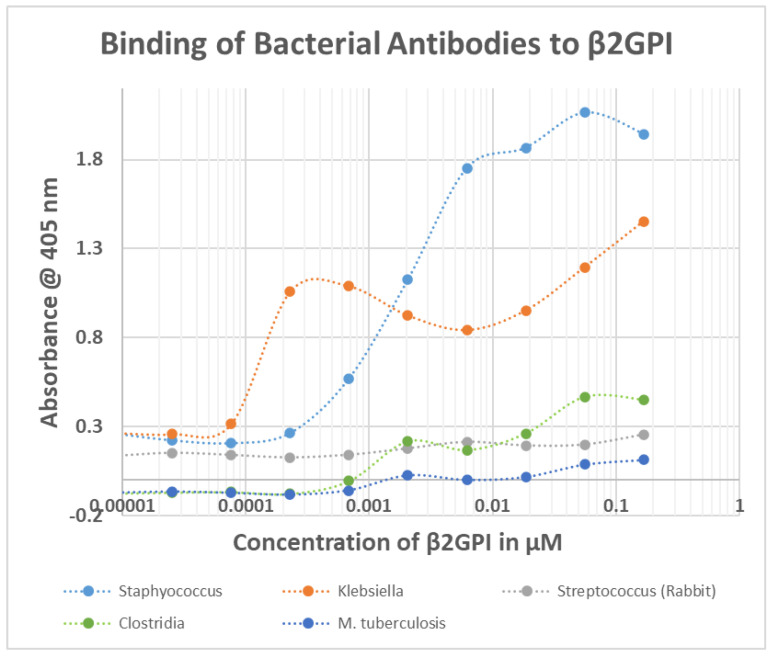
Binding of bacterial antibodies to beta 2 glycoprotein I (β2GPI). These are the results of a single experiment, which is why there are no error bars. Results were confirmed in by separate replication.

**Figure 9 ijms-23-11500-f009:**
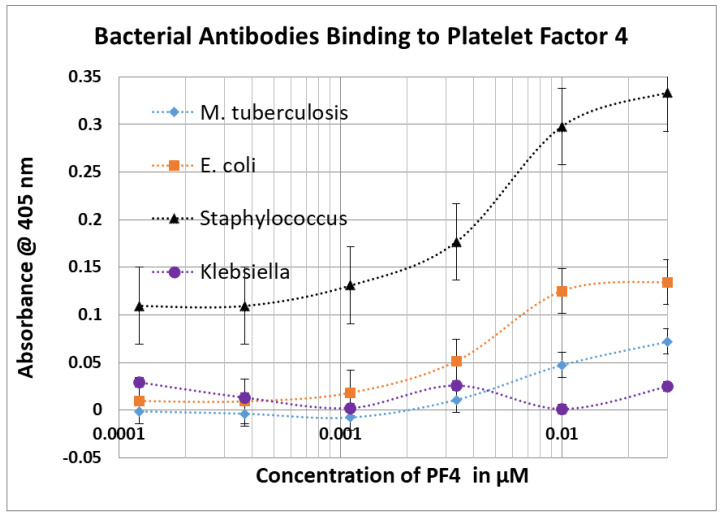
ELISA results showing binding of bacterial antibodies to platelet factor 4 (PF4). Experiments were run in duplicate.

**Figure 10 ijms-23-11500-f010:**
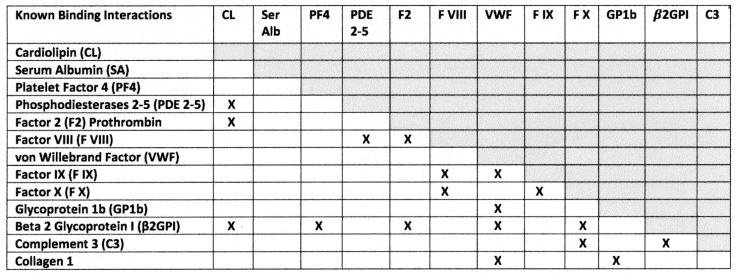
Summary of known binding interactions between proteins involved in the coagulation pathway. See text (Section 2.4) for references. CL = cardiolipin; Ser Alb = serum albumin; PF4 = platelet factor 4; PDE = phosphodiesterase; F2 = factor 2; FVIII = factor VIII; vWF = von Willebrand Factor; FIX = factor IX; FX = factor X; GP1b = glycoprotein 1b; *β*2GPI = beta 2 glycoprotein I; C3 = complement C3.

**Figure 11 ijms-23-11500-f011:**
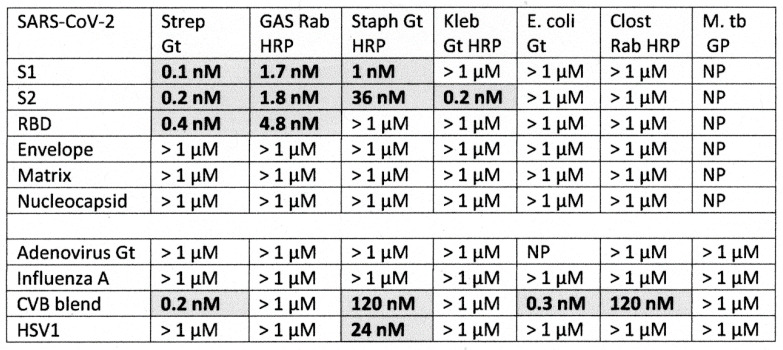
Results of double-antibody ELISA (DA-ELISA) experiments testing whether polyclonal antibodies against viruses recognized polyclonal antibodies against bacteria as “antigens”. Significant binding affinities (Kd 100 nM or smaller) are bolded against a grey background. NP = not possible to run the experiment because it was found that anti-rabbit horse-radish-peroxidase-labelled (HRP) antibodies recognized guinea pig antibodies and anti-guinea-pig-HRP antibodies recognized rabbit antibodies. S1, S2, RBD, Envelope, Matrix and Nucleocapsid are proteins of SARS-CoV-2; Gt = goat antibody; CVB blend is a mixture of coxsackievirus type B antibodies; HSV1 = human herpes simplex type 1 antibodies.

**Figure 12 ijms-23-11500-f012:**
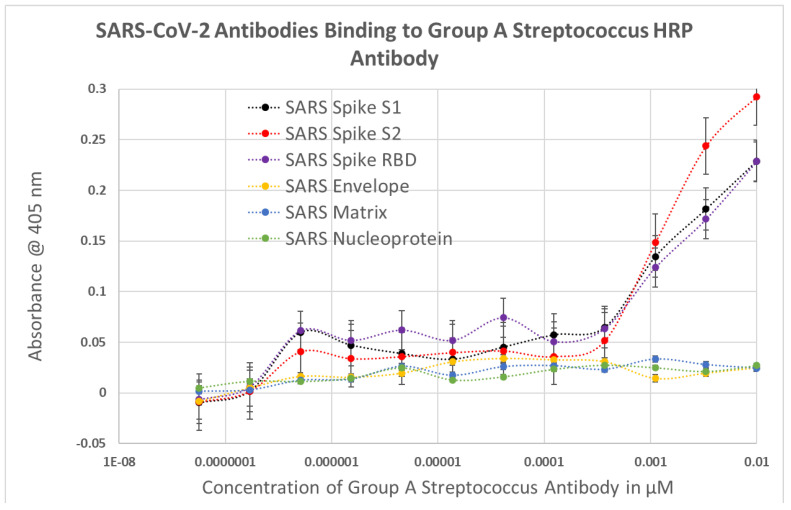
Results of DA-ELISA experiments testing for SARS-CoV-2 antibodies binding to *Group A Streptococcus* antibodies. Because both the virus and bacterial antibodies are polyclonal, multiple sets of complementary interactions are possible. Thus, while there is very clear evidence of binding between the Streptococcal antibodies and all three antibodies against SARS-CoV-2 spike region proteins at high concentrations of antibody, there also appears to be higher affinity binding at much lower concentrations of antibody (to the left). Only the high-concentration binding has been incorporated into Figure 11. S1, S2, RBD, Envelope, Matrix and Nucleocapsid are proteins of SARS-CoV-2.

**Figure 13 ijms-23-11500-f013:**
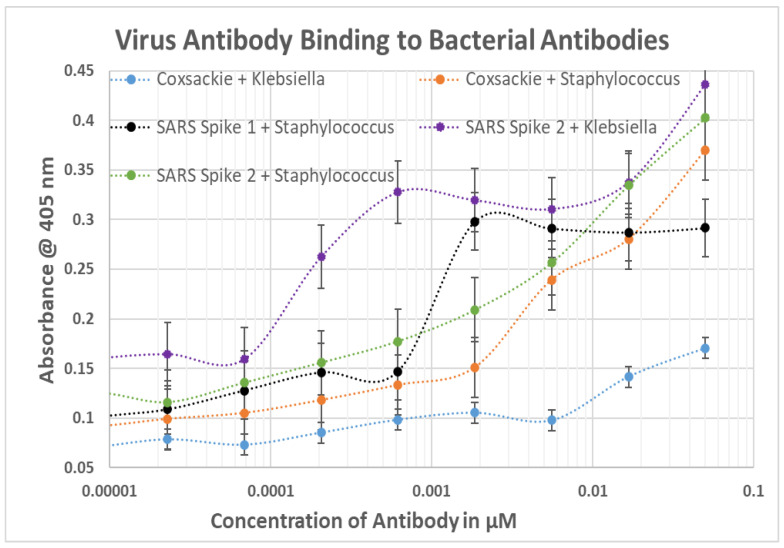
Results of DA-ELISA experiments testing for various virus antibodies binding to various bacterial antibodies. Because both the virus and bacterial antibodies are polyclonal, multiple sets of complementary interactions are possible and the resulting curves are often not the simple S-shaped ones that are normally associated with binding curves. SARS = SARS-CoV-2.

**Figure 14 ijms-23-11500-f014:**
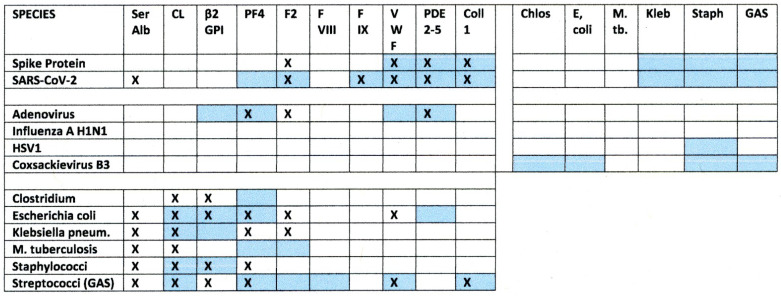
Summary of the Results. X’s represent similarities predicted by similarity searching summarized above in Figure 1 while the blue color represents experimentally significant binding reported in the Results section. Significance of antibody binding to proteins was determined as described in Table 2 as a function of the antibody affinity. No predictions have previously been made concerning the likelihood of an antibody induced by one microbe recognizing as antigenic an antibody raised against a different microbe, so there are no X’s in that portion of this Figure. Significant binding of one antibody to another was set at a Kd of 100 nM or less assuming that the concentration of any one antibody idiotype is unlikely to exceed 1 nM and therefore antibodies with less than a mutual binding constant of 100 nM are extremely unlikely to encounter each other in sufficient quantities to produce circulating immune complexes. CL = cardiolipin; Ser Alb = human serum albumin; β2GP = beta 2 glycoprotein; C1q, C3, C4 and C5 are complement proteins 1q, 3, 4 and 5; PDE = phosphodiesterase; F2, FVIII, FIX and FX are blood coagulation factors; VWF is von Willebrand’s factor; Coll 1 is collagen type 1. Bacterial abbreviations refer to the bacteria listed in the left-most column.

**Figure 15 ijms-23-11500-f015:**
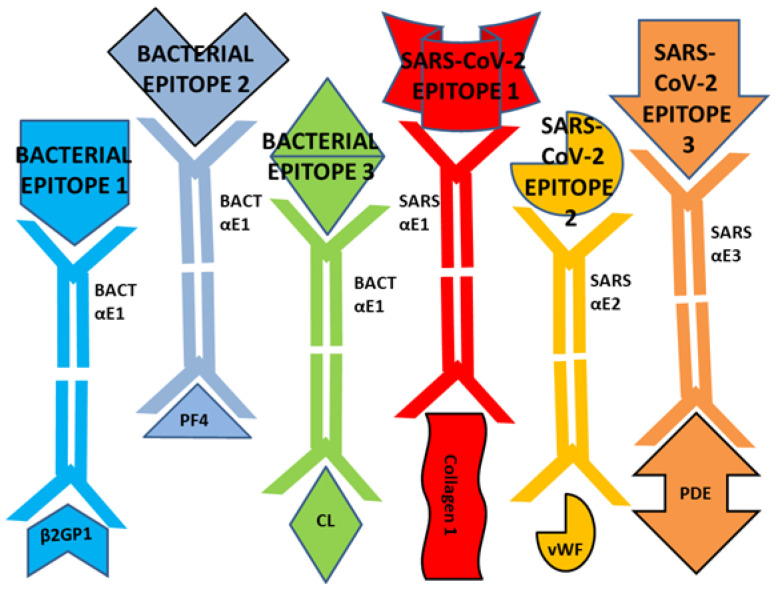
Antibodies against bacterial epitopes (BACT αE1, αE2, αE3)—in blues and greens- and antibodies against viral epitopes (SARS αE1, SARS αE2, SARS αE3)—in oranges and reds—cross-react with human blood proteins due to mimicry between their proteins: β2GPI = β2glycoprotein1; PF4 = platelet factor 4; CL = cardiolipin; vWF = von Willebrand Factor; PDE = phosphodiesterase.

**Figure 16 ijms-23-11500-f016:**
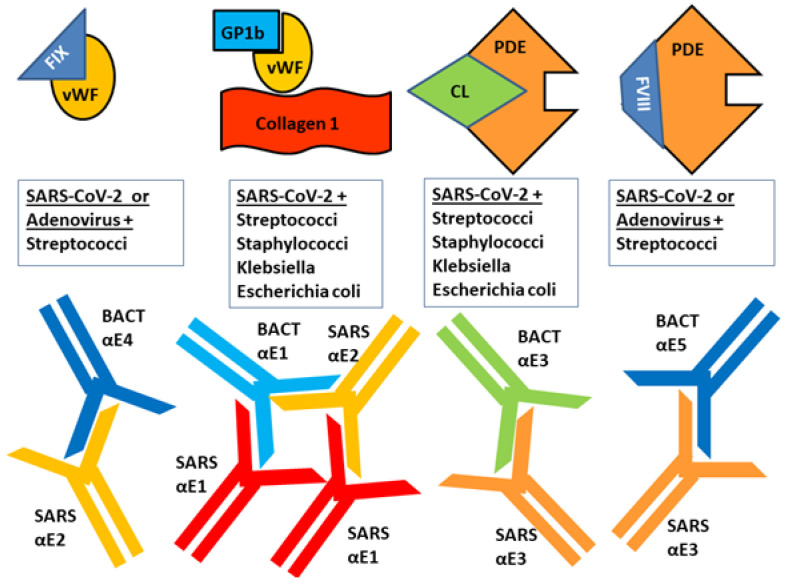
Because many of the blood proteins targeted in COVID-19 coagulopathies are molecularly complementary to each other, they will present complementary epitopes. Microbial antibodies against SARS-CoV-2 epitopes (SARS aE1, SARS aE2, SARS aE3) may then form circulating immune complexes by binding to bacterial antibodies (BACT aE2, BACT aE3, etc.) that can cross-react with the complementary epitope. GP1b = platelet glycoprotein 1b; FVIII = blood coagulation factor VIII. See Figure 16 for other abbreviations.

**Figure 17 ijms-23-11500-f017:**
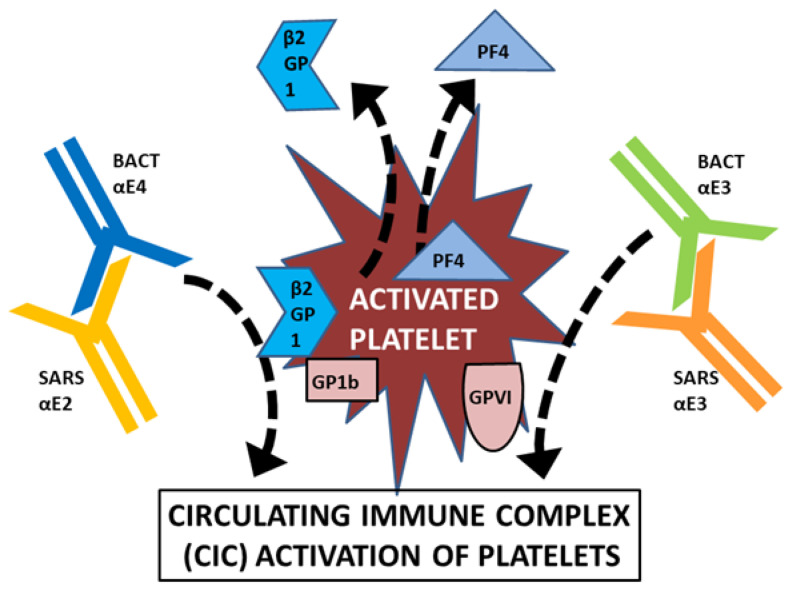
Circulating immune complexes formed from complementary bacterial (BACT) and SARS-CoV-2 virus (SARS) antibody idiotypes, particularly those involving PF4-reactive antibodies, will activate platelets leading to coagulation initiation. GP1b = platelet glycoprotein 1b; GPVI = platelet glycoprotein VI. See Figure 16 for other abbreviations.

**Figure 18 ijms-23-11500-f018:**
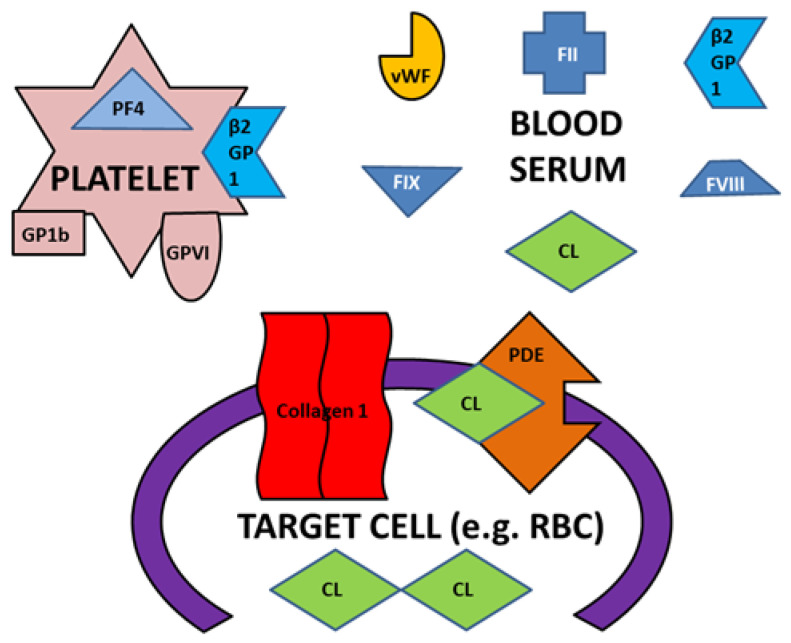
Some of the key blood proteins that were the focus of the present study and their relationships to platelets, red blood cells (RBC) and blood serum components. β2GPI = β2glycoprotein1; PF4 = platelet factor 4; CL = cardiolipin; vWF = von Willebrand Factor; PDE = phosphodiesterase; F2 = blood coagulation factor II (prothromgin); FVIII = blood coagulation factor VIII; FIX = blood coagulation factor IX; GP1b = platelet glycopro-tein 1b; GPVI = platelet glycoprotein VI.

**Figure 19 ijms-23-11500-f019:**
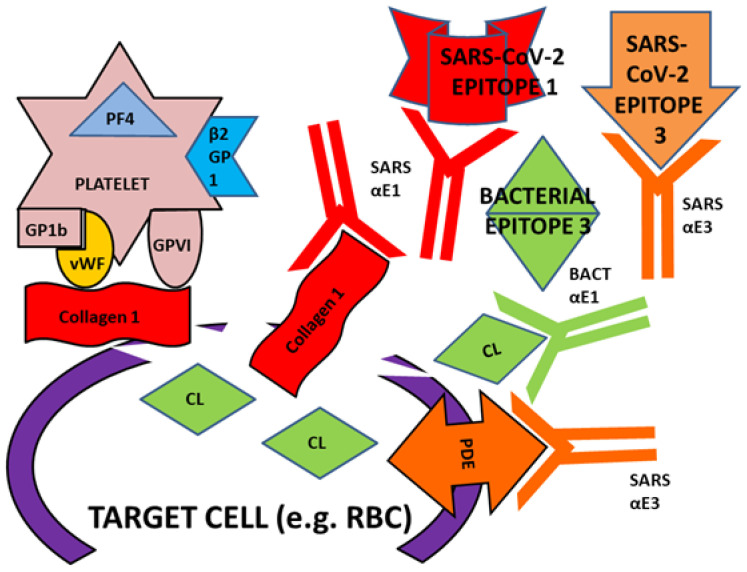
Initiation of platelet activation involves exposure of collagen, which may be achieved by a direct attack on collagen itself, or on other cellular proteins such as phosphodiesterases (PDE), cardiolipin (CL), or a CL-PDE complex (not shown). Collagen exposure results in von Willebrand Factor (vWF) binding to platelet glycoprotein 1b in concert wit platelet glycoprotein VI, which activates platelets to begin the blood coagulation process. See Figure 16 for the rest of the abbreviations.

**Figure 20 ijms-23-11500-f020:**
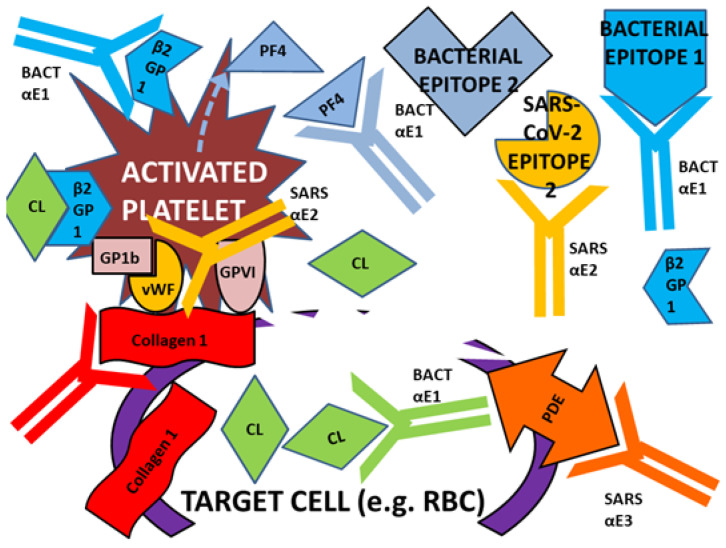
Simplified illustration of the many possible cross-reactive antibody targets that may result from epitope similarities shared by bacterial, viral and blood proteins. See Figure 20 for further information about the initiation of some of these reactions and Figure 16 for abbreviations.

**Table 1 ijms-23-11500-t001:** Proteins involved in blood coagulation that are targets of autoantibodies in COVID-19 patients and the functions and interactions among these proteins.

Protein	Function	Interactions
Coagulation Proteins		
Factor 2 (F2) (Phosphatidylserine/prothrombin)	When cleaved to thrombin, converts fibrinogen to fibrin (i.e., clot formation)	Activated by FX
Factor VIII (FVIII)	Cofactor for FX activation by Factor Xa	Forms complex with FX
Factor IX (FIX)	Activated by tissue factor (TF) and Factor VIIa to activate FVIII	Forms complex with FVIII; complex forms on phospholipid scaffolds on platelets
Factor X (FX)	Cleaves prothrombin (F2) to thrombin	Forms complex with F2
Factor Xa (FXa)	Activated by tissue factor (TF) and Factor VIIa; in concert with FXIII, activates FX	Complex forms on phospholipid scaffolds on platelets
Von Willebrand Factor (vWF)	Facilitates platelet aggregation to form temporary clots	Binds to FIX and GP1a on activated platelets and collagens exposed on damaged cells
Platelet Proteins		
Glycoprotein 1b (GP1b)	Activates platelet aggregation	Binds Factor V and FIX to the surfaces of platelets to activate them
Lupus anticoagulant (LA)	Antibodies against PL, PLBP, and PDE, inactivating them	Bind to PL and PLBP
Phosphodiesterases (PDE)	Regulate platelet activation by regulating phospholipid-binding proteins	Modify phospholipid-binding proteins such as β2GPI
Beta-2 Glycoprotein I (β2GPI)	Phospholipid binding protein that acts as an anti-coagulant	Bind to phospholipids such as CL to inactivate them
Cardiolipin (CL)(a phospholipid)	Promote coagulation by acting as scaffolds for blood factor binding	Necessary for FVIII and FXa activity; bind to phospholipid binding proteins such as β2GPI
Platelet Factor 4 (PF4)	Released by activated platelets to neutralize heparin permitting coagulation	Binds to and inactivates heparin-like molecules
Tissue Proteins		
Collagens	Extracellular cell matrix proteins involved in cell adhesion and integrity	vWF binds to exposed collagens to initiate FIX and GP1a platelet binding

**Table 2 ijms-23-11500-t002:** The Kd range of antibodies found in experiments reported here compared with the serum or platelet concentration of the same proteins. In order for an antibody to bind up significant amounts of a protein to interfere with protein function or to create circulating immune complexes, the binding constant of the antibody must be smaller than or on the same order as the concentration of the protein in blood serum or within platelets. Conc = concentration; n/r = not relevant.

Protein	Significant Kd Found Here	Conc. in Serum	Conc. in Platelets	Source
Cardiolipin	400 nM–2 µM	10 µM		[65]
β2 Glycoprotein I	2–200 nM	3 µM		[66]
Platelet Factor 4	5–100 nM	0.4 nM	400 nM	[67]
Serum Albumin	NONE	500 µM		[68]
Factor II (Prothrombin)	2–200 nM	2 µM		[69]
Factor VIII	0.4–0.5 nM	1 nM		[70]
Factor IX	0.3–0.9 nM	90 nM		[71]
von Willebrand Factor	10–30 nM	2 µM		[72]
Phosphodiesterases	8–40 nM	12 µM	7 µM	[73,74]
Collagen 1	4–8 nM	n/r	n/r	

**Table 3 ijms-23-11500-t003:** Antibodies utilized in this study.

Antibody Target	Species	Supplier	Product #
Adenovirus	Goat	Millipore (Burlington, MA, USA)	AB1056
*Clostridia*	Rabbit	Invitrogen (Waltham, MA, USA)	PA1-7210
*Clostridia*	Rabbit	Invitrogen	PA1-7210
*Clostridium* sp. HRP	Rabbit	US Biological (Swampscott, MA, USA)	C5853-25C
Coxsackie Virus B-Blend	Mouse	Millipore	MAB9410
*Escherichia coli*	Goat	Abcam (Cambridge, England)	AB13627
Goat Anti-Mouse IgG HRP	Goat	Sigma-Aldrich (Burlington, MA, USA)	A9917
Goat Anti-Rabbit IgG HRP	Goat	Invitrogen	65-6120
Herpes Simplex Virus Type 1	Goat	Invitrogen	PA1-7493
Influenza A HRP	Goat	Biodesign International (Palo Alto, CA, USA)	B65243G
*Klebsiella pneumoniae* HRP	Rabbit	Invitrogen	PA1-73176
*Mycobacterium tuberculosis*	Rabbit	ABD Serotec (Kidlington, UK)	OBT0947
*Mycobacterium tuberculosis*	Guinea Pig	MyBioSource (San Diego, CA, USA)	MBS315001
Rabbit Anti-Goat IgG HRP	Rabbit	Millipore	AP106P
Rabbit Anti-Guinea Pig HRP	Rabbit	abcam	AB6771
SARS-CoV-2 Envelope protein	Rabbit	Invitrogen	PA1-41158
SARS-CoV-2 Matrix protein	Rabbit	Invitrogen	PA1-41160
SARS-CoV-2 Nucleocapsid	Rabbit	Invitrogen	PA5-116894
SARS-CoV-2 Spike Protein RBD	Rabbit	Millipore	ABF1064
SARS-CoV-2 Spike Protein S1	Rabbit	Millipore	ABF1065
SARS-CoV-2 Spike Protein S1	Rabbit	Invitrogen	PA5-116916
SARS-CoV-2 Spike Protein S2	Rabbit	Millipore	ABF1063
*Staphylococcus aureus*	Rabbit	Invitrogen	PA1-7246
*Staphylococcus aureus* HRP	Rabbit	Invitrogen	PA1-73173
*Streptococcus Group A*	Goat	Invitrogen	PA1-7249
*Streptococcus Group A* HRP	Rabbit	Acris Antibodies (Herford, Germany)	BP2026HRP
*Streptococcus pneumoniae*	Rabbit	Biodesign International	B65831R
*Streptococcus pneumoniae*	Rabbit	Invitrogen	PA1-7259

**Table 4 ijms-23-11500-t004:** Proteins used in this study.

Protein	Species	Supplier	Product #	Purity
Beta 2 glycoprotein I	Human	Prolytix	B2G1-0001	>95% by SDS-PAGE
Cardiolipin sodium salt	Bovine	Sigma-Aldrich	C0563	≥97% (TLC)
Coagulation Factor VIII (rDNA)	Human	Sigma-Aldrich	H0920000	>99%, European Pharmacopoeia Reference Standard
Coagulation Factor IX (rDNA)	Human	Sigma-Aldrich	Y0001659	>99%, European Pharmacopoeia Reference Standard
Collagen 1	Human	Sigma-Aldrich	C7774	>95% SDS Electrophoresis
Platelet Factor 4	Murine	Sigma-Aldrich	SRP3231	≥98% (SDS-PAGE), ≥98% (HPLC)
Prothrombin	Human	Sigma-Aldrich	539515	>95% (SDS-PAGE)
Phosphodiesterase II	Bovine	Sigma-Aldrich	P9041	≥5.0 units/mg protein
Serum Albumin	Human	Sigma-Aldrich	A1653	≥96% (agarose gel electrophoresis)
von Willebrand Factor	Human	Sigma-Aldrich	681300	≥95% (SDS-PAGE)

## Data Availability

Data not presented in full in the body of this paper can be obtained by writing to the corresponding author.

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
