# Peer review of "Complementary Sets of Autoantibodies Induced by SARS-CoV-2, Adenovirus and Bacterial Antigens Cross-React with Human Blood Protein Antigens in COVID-19 Coagulopathies"

_ijms, 2022, doi:10.3390/ijms231911500_

Round 1

Reviewer 1 Report

This is an interestinly good experimental paper. There is, however, much room for improvement.

1) Many siceintists  may find it diffiicult to read as there needs to be a better summary of the antigenic proteins and  their functions. Many scientists will attemp to relate the function of the proteins to the virus antiboies and may find it hard to do so without deeper reading into the paper. I suggest an extra table with the names of the proteins and thir functions/description.

2) I was not able to find any description of the statistical methods used or the statistiical package used to do the calculations.

3) I was trying to figure out what R2  on the supplementary tables is until I looked at the fnumbers and realized it is the coefficient of determination (r2) with r representing the correlation coefficient. It is ususally wirtten as r2 with the 2 as a superscript, not R2. There is no description on how the r2 was calculated. Was it calculated using multivariiate analysis? 

2)

Author Response

REVIEWER 1

Open Review

English language and style

( ) Extensive editing of English language and style required
( ) Moderate English changes required
(x) English language and style are fine/minor spell check required
( ) I don't feel qualified to judge about the English language and style

Yes

Can be improved

Must be improved

Not applicable

Does the introduction provide sufficient background and include all relevant references?

( )

( )

(x)

( )

Are all the cited references relevant to the research?

(x)

( )

( )

( )

Is the research design appropriate?

(x)

( )

( )

( )

Are the methods adequately described?

( )

( )

(x)

( )

Are the results clearly presented?

( )

( )

(x)

( )

Are the conclusions supported by the results?

(x)

( )

( )

( )

Comments and Suggestions for Authors

This is an interestinly good experimental paper. There is, however, much room for improvement.

1) Many siceintists  may find it diffiicult to read as there needs to be a better summary of the antigenic proteins and  their functions. Many scientists will attemp to relate the function of the proteins to the virus antiboies and may find it hard to do so without deeper reading into the paper. I suggest an extra table with the names of the proteins and thir functions/description.

Good suggestion, thank you! The new Table has been added and a new paragraph more clearly explaining the protein functions and interactions.

2) I was not able to find any description of the statistical methods used or the statistiical package used to do the calculations.

Our oversight! Now added to Methods section.

3) I was trying to figure out what R2  on the supplementary tables is until I looked at the numbers and realized it is the coefficient of determination (r2) with r representing the correlation coefficient. It is ususally wirtten as r2 with the 2 as a superscript, not R2. There is no description on how the r2 was calculated. Was it calculated using multivariiate analysis? 

This has been corrected and the calculation (basic nonlinear exponential curve fitting equation at MyCurveFit.com) provided in the Methods section along with the correction addressed in 2) above. We, however, prefer R2 to r2, the former being the more common usage.

Submission Date

31 August 2022

Date of this review

09 Sep 2022 02:25:29

Reviewer 2 Report

The work that was done by Root-Bernstein et. al. studied the cross-reactivity between antibodies for various pathogens and major human proteins. The serious limitation of the presented work is that antibodies that commercially available polyclonal antibodies that were used for cross-reactivity studies may not represent the repertoire of antibodies that are produced in humans against a particular infection. To resolve this issue, we ask the authors to provide additional evidence that substitutions of the natural human-derived polyclonal repertoires against the antigen have the same cross-reactive properties as the commercially available polyclones.

Other comments

1)      Figure 1 was labeled as Table 1 in the introduction section. Also, the criteria for “high-similarity” were not stated for Table 1 data

2)      In figures with ELISA experiments, the dots are connected via solid splines. These lines should be replaced with dashed ones as no real measurements were done between these dots

3)      The results for polyclones from different vendors are not distinguished in Figure 2. Were the affinity measurements found to be the same for polyclones from different vendors?

Author Response

REVIEWER 2

Open Review

English language and style

( ) Extensive editing of English language and style required
( ) Moderate English changes required
( ) English language and style are fine/minor spell check required
(x) I don't feel qualified to judge about the English language and style

Yes

Can be improved

Must be improved

Not applicable

Does the introduction provide sufficient background and include all relevant references?

( )

(x)

( )

( )

Are all the cited references relevant to the research?

( )

(x)

( )

( )

Is the research design appropriate?

( )

( )

(x)

( )

Are the methods adequately described?

( )

(x)

( )

( )

Are the results clearly presented?

( )

( )

(x)

( )

Are the conclusions supported by the results?

( )

( )

(x)

( )

Comments and Suggestions for Authors

The work that was done by Root-Bernstein et. al. studied the cross-reactivity between antibodies for various pathogens and major human proteins. The serious limitation of the presented work is that antibodies that commercially available polyclonal antibodies that were used for cross-reactivity studies may not represent the repertoire of antibodies that are produced in humans against a particular infection. To resolve this issue, we ask the authors to provide additional evidence that substitutions of the natural human-derived polyclonal repertoires against the antigen have the same cross-reactive properties as the commercially available polyclones.

We completely agree with the Reviewer that validating our results with human polyclonal antibodies or sera would be a very good idea (as we already noted in our paragraph on “limitations” in the Methods section, which we have now expanded with the underlined sentence). However, we have encountered five problems that prevent us from doing so: 1) we have not been able to identify a commercial supplier that provides information about whether their sera come from COVID-19 patients with or without coagulopathies (we have written to several); 2) as noted in the original Methods section, because we are trying to differentiate cross-reactivities due to SARS-CoV-2, adenovirus, etc. from those due to bacterial infections, we cannot use sera from human patients who may have antibodies to any or all of these, since this would simply provide confusing and uninformative results; 3) we are an unusual Physiology Department in having no association with a hospital and so have no direct access to COVID-19 blood samples through colleagues; 4) our laboratory does not have permission to use human subjects or materials; 5) our laboratory is not approved (BCL2 or higher) to use materials that may contain infectious agents. We cannot, therefore, carry out the requested experiments, even if it were possible to obtain polyclonal antibodies specifically against single infectious agents, and to obtain human use approval, upgrade our lab, etc. would take many months in any case. If the Reviewer has access to the appropriate materials and can do such experiments, or knows an appropriate individual or commercial source who can supply the appropriate materials or do such experiments, we would be happy to collaborate!

Other comments

1)      Figure 1 was labeled as Table 1 in the introduction section. Also, the criteria for “high-similarity” were not stated for Table 1 data

Fixed.

2)      In figures with ELISA experiments, the dots are connected via solid splines. These lines should be replaced with dashed ones as no real measurements were done between these dots

 Fixed.

3)      The results for polyclones from different vendors are not distinguished in Figure 2. Were the affinity measurements found to be the same for polyclones from different vendors?

      They were the same. That point has now been made clear in the Methods section as well as the Figure caption.

Submission Date

31 August 2022

Date of this review

12 Sep 2022 08:33:54

Round 2

Reviewer 1 Report

Improvements seen.

Author Response

Thank  you!

Reviewer 2 Report

Unfortunately, in the revised version of the manuscript, the authors did not provide any evidence (even from previously published research) that the replacement of natural antibodies with commercial ones is valid. Instead, they just assumed that the commercially available polyclonal antibodies should have the same cross-reactivity pattern as natural ones, and used this statement as a known fact. As the source and methodology of the production of various antibodies that were used in the study remain obscure, the extrapolation of the results onto the physiological level seems to be a bit speculative. Considering the fact that the authors did not modify all the figures as was suggested (although they claimed that they did, figure 12 is still the same), I recommend performing major revisions of their manuscript.

Author Response

Open Review

(x) I would not like to sign my review report

( ) I would like to sign my review report

English language and style

( ) Extensive editing of English language and style required

( ) Moderate English changes required

( ) English language and style are fine/minor spell check required

(x) I don't feel qualified to judge about the English language and style

Comments and Suggestions for Authors

Unfortunately, in the revised version of the manuscript, the authors did not provide any evidence (even from previously published research) that the replacement of natural antibodies with commercial ones is valid. Instead, they just assumed that the commercially available polyclonal antibodies should have the same cross-reactivity pattern as natural ones, and used this statement as a known fact.

Respectfully, the Reviewer is incorrect. The Discussion, lines 554-622, provided, from the very first submission, a detailed discussion over three paragraphs of how our results compare with human antibody and sera studies. We have now added a number of phrases to emphasize the nature of the comparisons and the fact that, in the vast majority of cases, our rodent antibody data are consistent with human antibody or sera studies. These additions are indicated in red font, underlined, for ease of identification.  We also note that we did cite, and describe from the very first submission, the study by Vojdani, et al., that explicitly compared rodent and human antibody experiments, so we find the Reviewer’s comment that we have not even cited previously published research to be, frankly, incomprehensible.

As the source and methodology of the production of various antibodies that were used in the study remain obscure,

Is the Reviewer serious?! We provide the suppliers and the specific antibody product numbers for every antibody that we used in the Methods section. ANYONE can go online and access the suppliers’ detailed descriptions of what antigens were used to inoculate what animals and how the resulting antibodies were validated, and often tested for cross-reactivity.

 the extrapolation of the results onto the physiological level seems to be a bit speculative.

Speculative seems like a rather condescending term to use, but yes, we have an ENTIRE PARAGRAPH about the limitations of our approach as the last paragraph in our Methods sections, which has been there from the very first submission. In light of the fact that our results are consistent with human studies (paragraph above), we believe that our data goes a bit beyond “speculative”. Additionally, please recall that we have performed peer-reviewed, published studies on probable similarities between SARS-CoV-2, other viruses and bacteria with human blood proteins and that this study was designed to test those predictions. In fact, as we discuss in a paragraph in the Discussion, the results we report here are highly consistent with the theoretical predictions. Thus, similarity searches, rodent antibodies and human sera are all consistent. That doesn’t mean that additional tests are not needed, but it certainly goes beyond mere speculation.

Considering the fact that the authors did not modify all the figures as was suggested (although they claimed that they did, figure 12 is still the same), I recommend performing major revisions of their manuscript.

Our apologies for missing one of the figures. It is now corrected.